# Effects of the Level and Composition of Concentrate Supplements before Breeding and in Early Gestation on Production of Different Hair Sheep Breeds

**DOI:** 10.3390/ani13050814

**Published:** 2023-02-23

**Authors:** Farida Belkasmi, Amlan Kumar Patra, Raquel Vasconcelos Lourencon, Ryszard Puchala, Lionel James Dawson, Luana Paula dos Santos Ribeiro, Fabiola Encinas, Arthur Louis Goetsch

**Affiliations:** 1American Institute for Goat Research, Langston University, Langston, OK 73050, USA; 2Department of Agriculture Sciences, University Mohamed El Bachir El Ibrahimi, El Anasser 34030, Bordj Bou Arreridj, Algeria; 3College of Veterinary Medicine, Oklahoma State University, Stillwater, OK 74078, USA

**Keywords:** hair sheep, reproductive performance, supplementation, feed intake, body condition score

## Abstract

**Simple Summary:**

The level of supplementation of low-quality forage diets before and in the early gestation period may influence the performance of different hair sheep breeds. Dorper, Katahdin, and St. Croix sheep consumed wheat straw ad libitum supplemented with 0.15% initial body weight (BW) of soybean meal or a 1:3 mixture of soybean meal and rolled corn at 1% BW for 162 days with 84 or 97 days pre-breeding. Supplement treatment by breed did not generally influence feed intake, BW, body condition and mass indexes, or reproductive performance. Although soybean meal given alone boosted straw intake, it resulted in lower total feed intake, litter size, and total litter birth weight compared with the higher level of supplementation. Body condition score and mass indexes and BW were affected by breed, but reproductive performance was not influenced. Therefore, hair sheep regardless of breed fed low-protein and high-fiber forage such as wheat straw should be supplemented at a higher level with consideration of the inclusion of a feedstuff(s) high in energy along with protein during these physiological periods.

**Abstract:**

Female hair sheep, 27 Dorper (DOR), 41 Katahdin (KAT), and 39 St. Croix (STC), were used to determine influences of the nutritional plane before breeding and in early gestation on feed intake, body weight, body condition score, body mass indexes, blood constituent concentrations, and reproductive performance. There were 35 multiparous and 72 primiparous sheep, with initial ages of 5.6 ± 0.25 years and 1.5 ± 0.01 years, respectively (average overall initial age of 2.8 ± 0.20 years). Wheat straw (4% crude protein; dry matter [DM] basis) was consumed ad libitum and supplemented with approximately 0.15% initial body weight (BW) of soybean meal (LS) or a 1:3 mixture of soybean meal and rolled corn at 1% BW (HS; DM). The supplementation period was 162 days, with the breeding of animals in two sets sequentially, with the pre-breeding period 84 and 97 days, and that after breeding began at 78 and 65 days, respectively. Wheat straw DM intake (1.75, 1.30, 1.57, 1.15, 1.80, and 1.38% BW; SEM = 0.112) was lower (*p* < 0.05), but average daily gain (−46, 42, −44, 70, −47, and 51 g for DOR-LS, DOR-HS, KAT-LS, KAT-HS, STC-LS, and STC-HS, respectively; SEM = 7.3) was greater (*p* < 0.05) for HS than LS treatment during the supplementation period. Additionally, changes in body condition score during the supplementation period (−0.61, 0.36, −0.53, 0.27, −0.39, and −0.18; SEM = 0.058), and changes in body mass index based on height at the withers and body length from the point of the shoulder to the pin bone (BW/[height × length], g/cm^2^) from 7 days before supplementation (day −7) to day 162 were −1.99, 0.07, −2.19, −0.55, −2.39, and 0.17 for DOR-LS, DOR-HS, KAT-LS, KAT-HS, STC-LS, and STC-HS, respectively; (SEM = 0.297) were affected by supplement treatment. All blood constituent concentrations and characteristics addressed varied with the day of sampling (−7, 14, 49, 73, and 162) as well as the interaction between the supplement treatment and the day (*p* < 0.05), with few effects of interactions involving breed. Birth rate (66.7, 93.5, 84.6, 95.5, 82.8, and 100.0; SEM = 9.83) and individual lamb birth weight (4.50, 4.61, 4.28, 3.98, 3.73, and 3.88 kg; SEM = 0.201) were not affected by supplement treatment (*p* = 0.063 and 0.787, respectively), although litter size (0.92, 1.21, 1.17, 1.86, 1.12, and 1.82; SEM = 0.221) and total litter birth weight (5.84, 5.74, 5.92, 7.52, 5.04, and 6.78 kg for DOR-LS, DOR-HS, KAT-LS, KAT-HS, STC-LS, and STC-HS, respectively; SEM = 0.529) were greater (*p* < 0.05) for HS than for LS. In conclusion, although there was some compensation in wheat straw intake for the different levels of supplementation, soybean meal given alone rather than with cereal grain adversely affected BW, BCS, BMI, and reproductive performance, the latter primarily through litter size but also via a trend for an effect on the birth rate. Hence, the supplementation of low-protein and high-fiber forage such as wheat straw should include a consideration of the inclusion of a feedstuff(s) high in energy in addition to nitrogen.

## 1. Introduction

The nutritional plane is of importance to small ruminants throughout the year, although there are specific periods and stages of production when it is critical. Additionally, of course, nutrient and energy intake at any given time influences needs later to achieve desired levels and efficiencies of production [1]. However, nutritional management and modeling for small ruminants are challenging because of very diverse farming practices, feeding habits, and environmental adaptation, including the unique resilience of different breeds to varying conditions in wide-ranging geographical areas [2].

Before and during the breeding period and early gestation are critical periods for sheep when feeding practices can have a marked influence on reproductive performance resulting from ovulation rate, embryo quality, and uterine environmental conditioning due to metabolic and endocrine balances [3,4,5]. Relatedly, ‘flushing’ for increased conception rate and litter size is a common practice in sheep production systems, particularly with basal dietary forage low to moderate in nutritional value or quality [6]. Additionally, proper nutrition and body condition score (BCS) during gestation influence the performance, health, reproduction, and metabolic responses of progeny [7,8].

The three most common breeds of hair sheep in the USA are Dorper (DOR), Katahdin (KAT), and St. Croix (STC) [9,10]. After a project in which the resilience of these breeds to stress factors was expected to increase in importance with anticipated future climate change [11,12,13], a study was conducted with some of these mature females used to ascertain the effects of different supplement treatments before and during the breeding period on body weight (BW) change and composition, reproductive performance, and a number of other variables such as BCS, body mass indexes (BMI), and blood constituent concentrations [14]. Wheat straw was consumed ad libitum and supplemented with 0.16% BW (dry matter [DM] basis) of soybean meal (SBM) or 0.8% BW (DM basis) of a 3:1 mixture of rolled corn and SBM before breeding for 2.25–2.50 months before breeding and the first 17 days of the breeding period. Although the intake of wheat straw was greater for the SBM than for the corn:SBM treatment, partially compensating for the different levels of supplement intake, changes in BW and body energy content, BCS, and BMI were greater for the corn:SBM vs. SBM treatment. However, reproductive performance was similar among supplement treatments, perhaps reflecting the mobilization of body energy stores to support reproductive performance. Moreover, there were no interactions in the reproductive performance between the hair sheep breed and the supplement treatment. Questions that arose from this study include whether results would differ with a slightly higher level of intake of the corn:SBM supplement as well as the potential impact of imposing treatments in the early gestation period. Therefore, the objectives of this study were to determine the effects of different levels of intake and compositions of supplemental concentrate before breeding and in early gestation on the production of Dorper, Katahdin, and St. Croix hair sheep consuming low-quality basal dietary forage.

## 2. Materials and Methods

### 2.1. Animals

The protocol for the experiment was approved by the Langston University Animal Care and Use Committee. The study occurred in the late summer of 2019 through lambing in the spring of 2020. There were 107 female hair sheep used, 27 DOR, 41 KAT, and 39 STC. The animals included 35 ewes that had previously given birth and 72 that had not, with initial ages of 5.6 ± 0.25 years and 1.5 ± 0.01 years, respectively (average overall initial age of 2.8 ± 0.20 years). The ewes were derived in the summer and early fall of 2015 from four climatic zones and regions of the USA, the Northwest (primarily Oregon), Midwest (parts of Iowa, Wisconsin, Minnesota, and Illinois), Southeast (primarily Florida), and central Texas. The animals were used in a series of trials from the fall of 2015 through the spring of 2018, described by Tadesse et al. [12,13] and Hussein et al. [11], evaluating resilience in stress factors projected to become increasingly important and prevalent with future climate change, namely, high heat load conditions, limited water availability, and restricted feed intake. After those studies, some animals were culled for various reasons such as old age, and others were designated for another experiment, with the ones remaining used in the current study. The ewe lambs were progeny from the ewes used in these previous studies.

### 2.2. Treatments and Housing

Before the experiment, sheep were vaccinated against clostridial organisms, and ones with a FAMACHA^©^ score greater than 3 were treated for internal parasites. The sheep were allocated to four groups per breed based on body weight (BW) and age, with 6 to 11 animals per group, and two supplement treatments were randomly assigned, with a table of random numbers, to the groups (Figure 1). Each group was housed in a pen with two sections, a 3.66 m × 3.66 m area within a building and a second with the same dimensions located outside but also covered by a roof. The floors were concrete and periodically cleaned. Water was available free-choice in automatic waterers and there was free access to salt blocks. Artificial lighting was provided from 07:00 to 16:00 h. There were two feed troughs (2.44 m and 1.22 m in length) per pen in the front area. There were six pens on one side of the facility and six on the other, with a hallway between the feed troughs and pens. There was one breed × supplement treatment group on each side of the facility.

Animals were weighed before supplement treatments were first imposed. Body weight was determined in the morning before refusals from the previous day were collected and new feedstuffs were dispensed. The supplement treatments were soybean meal fed at approximately 0.15% of initial BW and a mixture of 25% soybean meal and 75% rolled corn given at approximately 1% BW (dry matter basis; LS and HS, respectively). Coarsely ground wheat straw with crude protein (CP) content of 3.9% and neutral detergent fiber content (NDF) of 80.2% (Table 1) was consumed ad libitum and offered at approximately 120% of consumption on the preceding few days. After straw refusals were collected and weighed, supplements were fed at 08:00 h, which were completely consumed within a few minutes, and then wheat straw was offered. Wheat straw and supplement samples were collected weekly, ground to pass a 1 mm screen, and analyzed for dry matter (DM), ash [15], and CP by the Doumas method using a C/N analyzer (Leco TruMac CN, St. Joseph, MI, USA), and NDF with the use of heat stable amylase and containing residual ash [16], and acid detergent fiber (ADF) content in wheat straw and supplement and acid detergent lignin (ADL) content in wheat straw were analyzed using the filter bag technique (ANKOM Technology Corp., Fairport, NY, USA).

With day 1 considered the first day of imposing the supplement treatments, BW was determined on days −7, 14, 49, 73, 120, 162, 198, and 225, with periods 1, 2, 3, 4, 5, 6, and 7 considered 21, 35, 24, 47, 42, 36, and 27 days in length, respectively (Figure 2). The variables in the different periods were analyzed to capture dynamic changes affected by treatment in these periods with periods 1 to 5 for supplement treatment effects and periods 6 and 7 for previous supplements’ effects while the animals were on a similar nutritional plane. After the 162-day supplementation phase (i.e., end of period 5), there was an additional 63-day period that ended near the time when lambing began. The animals in periods 6 and 7 received the HS supplement treatment (i.e., 1% of initial BW) and were treated similarly in outside pens, and thereafter given a higher level of a concentrate-based supplement and free-choice access to moderate- to high-quality forage.

### 2.3. Breeding

One pen of animals of each treatment was assigned to 1 of 2 breeding sets. Four rams of each breed divided into 2 sets were used, which were previously subjected to and passed a breeding soundness examination. The 2 initial breeding periods were 7 days in length on days 84–90 and 97–103 for sets 1 and 2, respectively. Estrus synchronization was employed, with a controlled internal drug release (CIDR) vaginal device (EAZI-breed™ CIDR^®^, Pfizer Animal Health, Auckland, New Zealand) inserted on day 0, and 10 mg of PGF2α (Lutalyse, 10 mg dinoprost tromethamine i.m., Zoetis Animal Health, Parsippany-Troy Hills, NJ, USA) was intramuscularly injected 9 days later. On day 10, the CIDR was removed, at which time, 2 rams of each breed fitted with marking harnesses were introduced into a pen of the relevant breeding set. Estrus and breeding markings were checked twice daily at 08:00 and 16:00 h. Non-return to estrus was assumed when females were not marked during a second cycle with a ram at 15–18 days after an observed mating. Pregnancy was also diagnosed at 40 days after breeding using ultrasonography by ventral external examination with a 3.5 MHz linear-array transducer, and 12 open animals were removed from the study. Approximately 1 month before lambing, ewes were vaccinated again against clostridial organisms, and ones with FAMACHA© scores greater than 3 were treated for internal parasites. The ewes were moved back to the confinement facility approximately 1 week before lambing. The birth rate was the percentage of females exposed to rams that gave birth (95 of the 107 total, 30 of 35 multiparous, and 65 of 72 primiparous). Fecundity was based on the number of lambs born per female exposed to rams, and other reproduction measures were based on animals giving birth, which occurred in portable lambing pens. The gestation length was based on the last day of observed breeding and the date of birth.

### 2.4. Other Measures

Body condition score (BCS), as described by Ngwa et al. [17], was assessed by three or four individuals when BW was determined (i.e., on days −7, 14, 49, 73, 120, 162, 198, and 225). Linear measurements of height at the withers (Wither), length from the point of the shoulder to the hook bone (Hook) and pin bone (Pin), and circumference from heart girth (Heart) occurred on days −7, 14, 49, 73, and 162. Of the 13 body mass indexes (BMI) described by Liu et al. [18], the 4 most meaningful ones as recommended by Wang et al. [19] were calculated as noted below.
BMI–WH = BW/(Wither × Hook) [g/cm^2^](1)
BMI–WP = BW/(Wither × Pin) [g/cm^2^](2)
BMI–GH = BW/(Heart × Hook) [g/cm^2^](3)
BMI–GP = BW/(Heart × Pin) [g/cm^2^](4)

Blood samples (10 mL) were collected by jugular venipuncture into three tubes on days 14, 49, 73, and 162. There were two tubes used for plasma, one with sodium fluoride and potassium oxalate, and another with sodium heparin. A third tube without an anticoagulant was used to derive serum. Plasma and serum were harvested by centrifugation for 20 min at 3000× *g* and frozen at −20 °C. Plasma from the sodium fluoride and potassium oxalate tube was analyzed for glucose and lactate with a USI 2300 Plus Glucose & Lactate Analyzer (YSI Inc., Yellow Springs, OH, USA). Plasma from the sodium heparin tubes and (or) serum was analyzed for constituents such as nonesterified fatty acids (NEFA), triglycerides (TG), cholesterol, urea nitrogen (N), albumin, and total protein with a Vet Axcel^®^ Chemistry Analyzer (Alfa Wassermann Diagnostic Technologies, West Caldwell, NJ, USA) according to the manufacturer’s instructions. Total antioxidant capacity (TAC) in plasma was determined by measuring the ferric-reducing ability of plasma colorimetrically [20].

Heart rate (HR) was measured in set 1 animals the week preceding days 14, 49, 73, and 162 and on set 2 animals the following week. Heart rate was assessed to predict heat energy (HE) based on the ratio of HE to HR for each animal as employed in many other studies [21,22,23]. Heart rate measurement was as described by Puchala et al. [24,25]. Heart rate was determined for 24 h with animals in 4 pens at a time over a total period of 3 days. Sheep were fitted with stick-on ECG electrodes (Cleartrace, Utica, NY, USA) attached to the chest just behind and slightly below the left elbow and at the base of the jugular groove on the right side of the neck. Electrodes were secured to the skin with a 5 cm-wide elastic bandage (Henry Schein, Melville, NY, USA) and animal tag cement (Ruscoe, Akron, OH, USA). There was the use of ECG snap connecting leads (Bioconnect, San Diego, CA, USA) to connect electrodes to T61coded transmitters (Polar, Lake Success, NY, USA). Human S610 HR (Polar) monitors with wireless connection to the transmitters were used to collect HR data at a 1 min interval. Heart rate data were analyzed using Polar Precision Performance SW software.

For the HE to HR ratio, on days 80 to 95, animals were cycled in groups of 5 to 7 into a room with metabolism cages fitted with headboxes of a respiration calorimetry system for 1 day. Emissions of methane and carbon dioxide and oxygen consumption were measured with an indirect, open-circuit respiration calorimetry system (Sable Systems International, North Las Vegas, NV, USA) as described by Puchala et al. [24,25]. Oxygen concentration was analyzed using a fuel cell FC-1B oxygen analyzer (Sable Systems International), and methane and carbon dioxide concentrations were measured with infrared analyzers (CA-1B for carbon dioxide and MA-1 for methane; Sable Systems International). Before each measurement, analyzers were calibrated with reference gas mixtures. Heat energy was calculated from oxygen consumption and the production of carbon dioxide and methane according to the Brouwer [26] equation without the consideration of urinary nitrogen.

### 2.5. Statistical Analyses

Most data were analyzed with the MIXED procedure of SAS [27], with fixed effects of breed, supplement treatment, and their interaction and random effect of animal within breed and supplement. Because of the limited number of observations for multiparous animals, parity was not included in the model. However, for discussion purposes, reproduction data were also analyzed separately for multiparous and primiparous animals. The animal group or pen within the breed and supplement treatment was the experimental unit. Day was a repeated measure for the analysis of blood constituent concentrations. For the reproduction variables of birth rate and litter size, the analysis was with the GLIMMIX procedure of SAS [27]. Breed × supplement treatment means are presented regardless of the significance of the interaction. Differences among means were determined by the least significant difference with a protected F-test (*p* < 0.05). Pearson correlations (r) between BW, BCS, and BMI were determined using SAS [27].

## 3. Results

### 3.1. Feed Intake, BW, and ADG

The chemical composition of wheat straw and supplements (Table 1) was similar to that in the earlier experiment of Lourencon et al. [14]. The level of supplement DM intake in g/day and % BW was greater for High than for Low in all periods and overall (*p* < 0.05; Table 2). The intake of wheat straw DM in g/day and % BW was greater for LS vs. HS in each period (*p* < 0.05) except for period 1 (*p* > 0.05) when there was no compensation in wheat straw intake for the higher level of intake of HS. Conversely, there was partial compensation in periods 2 and 3 and overall, with greater total DM intake in % BW for HS than for LS (*p* < 0.05). In periods 4 and 5, total DM intake in % BW was similar between supplement treatments, as wheat straw intake was considerably higher in these periods for HS. Wheat straw DM intake was 0.29, 0.49, 0.61, 0.66, and 0.43% BW greater for LS vs. HS in periods 2, 3, 4, and 5 and overall, respectively. Likewise, relative to the wheat straw DM intake for HS, the intake for LS was 23.5, 37.5, 43.2, 49.9, and 33.7% greater in periods 2, 3, 4, and 5 and overall, respectively. Although an analysis was not conducted to determine the period effects because of factors such as different lengths and stages of production, it is germane to note that the total DM intake increased as the periods progressed and plateaued in period 4 (1.70, 1.96, 2.09, 2.27, and 2.19% BW in period 1, 2, 3, 4, and 5, respectively). A breed effect (*p* < 0.05) was noted for total DM intake in g/day in period 4 (DOR > STC) and for overall average supplement intake in % BW (DOR > KAT).

There were no interactions between breed and supplement treatment in BW or ADG (*p* > 0.05; Table 3). Body weight was numerically lower for STC than for DOR and KAT at most times and was lower than for KAT on day 49 and for DOR and KAT on day 225 (*p* < 0.05). Body weight was similar between supplement treatments on days −7, 14, and 49 (*p* > 0.05) but was greater for HS than for LS on subsequent days (*p* < 0.05). The values were 7.0, 7.8, 15.1, 11.2, and 12.0 kg and 13.6, 15.2, 31.3, 21.1, and 20.4% greater for HS than for LS on days 73, 120, 162, 198, and 225, respectively.

Average daily gain in periods 1–5 was similar among breeds (*p* > 0.05) except for a lower mean for STC vs. KAT in period 2 (*p* < 0.05; Table 3). Additionally, ADG by KAT in period 6, when all animals received HS and had access to a higher quality forage, was the lowest among the breeds (*p* < 0.05). Supplement treatment affected ADG in all periods (*p* < 0.05) except periods 7 and 6–7. The values averaged 93, 106, 123, 71, and 113 g greater for HS than for LS in periods 1, 2, 3, 4, and 5, respectively (*p* < 0.05). In period 6, ADG averaged 110 and 22 g for LS and HS, respectively (*p* < 0.5), with an 88 g difference. In period 7, ADG was similar between supplement treatments (*p* > 0.05), resulting in only a numerical difference in ADG of 39 g (*p* > 0.05) in periods 6–7. Overall, in periods 1–7, ADG averaged 61 g greater for HS vs. LS (*p* < 0.05; 9 and 70 g for LS and HS, respectively).

### 3.2. BCS and BMI

Body condition score was greatest among the breeds for KAT and greater for HS than for LS on days −7, 49, 73, 120, and 162 (*p* < 0.05; Table 4). On day 198, BCS was also greater for HS vs. LS (*p* < 0.05), although the BCS for DOR was greater than for KAT and STC (*p* < 0.05). There was an interaction between breed and supplement treatment in BCS on day 225 (*p* = 0.012), which was due in part to similar values for DOR with both LS and HS supplement treatments (*p* > 0.05) conversely to greater values for HS vs. LS for KAT and STC (*p* < 0.05). The overall BCS mean did not vary much among days, averaging 3.13, 3.19, 3.10, 3.22, 3.00, 3.03, 3.03, and 3.02 on days −7, 14, 49, 73, 120, 162, 198, and 225, respectively.

The only effect of breed on change in BCS without an interaction between breed and supplement treatment was for days 162–198, with means ranking (*p* < 0.05) KAT < STC < DOR (Table 4). There were breed by supplement treatment interactions (*p* < 0.05) in changes in BCS on days 120–162, 198–225, and 162–225, which were due to differences between supplement treatments for DOR (*p* < 0.05) but not KAT or STC. The main effect means of BCS change for supplement treatments were greater for HS vs. LS (*p* < 0.05) until 120 days, but thereafter, BCS change was similar (*p* > 0.05) between supplement treatments. The magnitudes of change in BCS between –7 and 162 days and 0 and 225 days were thus greater for HS vs. LS as well as between 162 and 225 days. 

There were differences (*p* < 0.05) among breeds in BMI on days −7, 14, 49, 73, and 162, and when not significant, there were tendencies for differences (*p* < 0.12; Table 5). In many instances, BMI was lowest among breeds for STC and similar among DOR and KAT, although in a small number of cases, the BMI for STC differed only from that of KAT. Supplement treatment did not affect BMI on day −7 as would be expected (*p* > 0.05), and this was also true for values on days 14 and 49. However, each of the four BMIs was greater for HS vs. LS on day 162 (*p* < 0.05), and there were similar differences (*p* < 0.05) or tendencies for BMI on day 73 (*p* < 0.12). There was only one case (BMI–WH between 73 and 162 days) in which change in BMI differed among breeds (*p* < 0.05; DOR < KAT), being similar otherwise (*p* > 0.05). In most time periods, the change in BMI was greater for HS vs. LS (*p* < 0.05) or tended to differ in this manner. The differences between supplement treatments in change during the entire time when different supplements were given were highly significant (*p* < 0.01) for each of the four BMIs. Differences for BMI–WH, BMI–WP, BMI–GH, and BMI–GP averaged 3.00, 2.09, 1.51, and 0.97 units greater for HS vs. LS, respectively. Although, it should be realized that BMI–WH and BMI–WP are based on Wither and BMI–GH and BMI–GP are based on Heart, with the latter being greater in magnitude (data not reported). Similarly, BMI–WP and BMI–GP are based on Pin, with BMI–WH and BMI–GH based on Hook, again with the magnitude of the latter being greater.

### 3.3. HR and HE

Breed did not impact HR or any measure of HE on day 14, 49, or 73 (*p* > 0.05; Table 6). Heart rate was greater for HS than for LS at each time. Heat energy was not affected by supplement treatment on day 14 (*p* > 0.05), but values in MJ/day were greater for HS on day 49 and 73 and in kJ/kg BW^0.75^ on day 49.

### 3.4. Blood Constituent Concentrations

All blood constituent concentrations were affected by day and the supplement treatment by day interaction (*p* < 0.05; Table 7). Moreover, there were some effects of breed and three breed by day interactions (*p* < 0.05). One variable, the concentration of triglycerides, was affected by a breed by supplement treatment interaction (*p* < 0.05).

The levels of total protein and albumin were lower on days 73 and 162 than earlier for LS, whereas for HS, the levels either decreased only slightly or remained relatively steady on these last 2 days of sampling (Table 8). Additionally, the level of total protein was lowest among the breeds for DOR (*p* < 0.05). The pattern of change with advancing time for the concentration of urea N also differed between supplement treatments. For LS, the level was highest among days for day −7, but for HS, the level increased and then declined as the day advanced. Additionally, the urea N concentration was greater for STC vs. KAT (*p* < 0.05), with an intermediate value for DOR (*p* > 0.05). The TG concentration was similar among days for LS, whereas for HS, the concentration was lowest among the days for day −7 and greatest for day 162 (*p* < 0.05). Breed by supplement treatment means for triglyceride concentration ranked are (*p* < 0.05) DOR-HS > KAT-HS > STC-HS, DOR-LS, KAT-LS, and STC-LS (*p* < 0.05). The concentration of cholesterol for LS increased and then decreased as the day advanced, and that for HS decreased and then increased to the same level as on day −7. The concentration of NEFA for HS was slightly greater on day −7 than on later days, although levels for LS increased markedly from day −7 to 14 and then declined slightly thereafter to levels above that initially on day −7.

The glucose concentration for LS decreased markedly from day −7 to days 14 and 49, increasing slightly thereafter (Table 8). Conversely, the concentration for HS decreased slightly from the value on day −7 and was steady thereafter. The concentration of glucose was relatively less for KAT at all times and did not markedly differ among days 14-162, whereas the values on day −7 for DOR and STC were high relative to those on subsequent sampling days. The lactate concentration for LS was considerably less on days 14 and 49 than on day −7, whereas values for HS differed among days relatively less. The lactate concentration for DOR and STC decreased and then increased with advancing time, but the level for KAT was fairly steady over the days. For HS, the TAC was lowest among days for day 162. Conversely, the values for LS on days 49 and 73 were greater than on days 14 and 162. The pattern of change in TAC with advancing time varied considerably among breeds, with values decreasing, increasing, and decreasing for DOR, increasing then decreasing for KAT, and lower on day 162 than on earlier days for STC.

### 3.5. Reproductive Performance

The birth rate was numerically (*p* = 0.063) greater for HS than for LS (96.3 vs. 78.0%; Table 9). The results were similar for the separate analysis of data from multiparous and primiparous animals, with supplement treatment *p*-values of 0.036 and 0.082, respectively. Litter size and fecundity were greater (*p* < 0.05) for HS vs. LS (1.69 vs. 1.37). For the supplement effect, the *p*-value of litter size for primiparous animals was similar (*p* = 0.047) to the *p*-value of overall litter size (*p* = 0.046), although the *p*-value (0.398) of litter size for multiparous animals was different. This difference between parity for the supplement effect appeared largely due to values for STC, with means of 1.00, 1.20, 1.50, 2.33, 2.00, and 1.92 for DOR-LS, DOR-HS, KAT-LS, KAT-HS, STC-LS, and STC-HS, respectively (SEM = 0.152). The only reproductive performance variable affected by breed was individual lamb birth weight (*p* = 0.035), being greater for DOR than for STC (4.56, 4.18, and 3.81 kg for DOR, KAT, and STC, respectively). The *p*-value for multiparous animals was similar at 0.062, and that for primiparous animals was slightly greater (0.169) because of numerical differences of lesser magnitude (4.43, 4.27, 4.25, 4.06, 3.66, and 3.92 kg for DOR-LS, DOR-HS, KAT-LS, KAT-HS, STC-LS, and STC-HS, respectively; SEM = 0.244). The total litter weight was greater (*p* = 0.049) as well for HS than for LS (6.68 vs. 5.60 kg). The *p*-value for primiparous animals was similar at 0.050. Conversely, the *p*-value for multiparous animals was 0.714, primarily because of values for STC and higher variability (5.03, 6.05, 7.85, 8.49, 7.52, and 7.10 kg for DOR-LS, DOR-HS, KAT-LS, KAT-HS, STC-LS, and STC-HS, respectively; SEM = 1.204). Gestation length was not influenced (*p* > 0.05) by breed or supplement treatment, and there were no breed by supplement treatment interactions for any reproductive performance variable (*p* > 0.05).

### 3.6. BMI Relationships

Correlation coefficients between BMI and BW were slightly greater for BMI–WH and BMI–WP on days 49, 73, and 162 and change from day −7 to day 162 relative to those for BMI–GH and BMI–GP, although the r values at earlier days were fairly similar (Table 10). The same was true for r values between BMI and BCS on day 162 and for change from day −7 to day 162, with similar values at earlier days among the four BMI. In all cases, the correlation coefficients between BMI–WH and BMI–WP and BW at different times and change in BW were greater than for BCS, and the same was true for BMI–GH and BMI–GP except for values for day 162 and change from day −7 to day 162.

## 4. Discussion

### 4.1. Feed Intake

For low-quality forages, digestive capacity including fermentation, the breakdown of feed particles, and digesta passage rate are important determinants of feed intake [28]. Low-quality forages are fermented in the rumen relatively slowly which can result in limited feed intake because of high ruminal NDF fill [29,30]. The supplementation of low-quality forages is necessary to support microbial growth in the rumen and improve fermentation. Usually, supplemental protein increases voluntary forage intake and digestibility when the forages contain less than 6 to 8% crude protein [31]. However, depending on numerous factors, adding a high-protein feedstuff(s) alone might not be sufficient.

The findings regarding feed intake by the three breeds of hair sheep in this study are fairly similar to those of Lourencon et al. [14], indicating that none was more capable than any other in achieving nutrient and energy intake adequate for higher levels of production with diets based on low-quality forage with supplement treatments varying in the level of feeding and chemical composition. Some of the findings regarding the effects of supplement treatments on feed intake also are similar to those of Lourencon et al. [14], although there were some interesting differences. In the current experiment, wheat straw DM intake in the first period, only 14 days in length, was similar between LS and HS, which is in contrast to greater wheat straw intake in % BW for LS in the study of Lourencon et al. [14] in each of the four periods. It seems that 14 days was not adequate to establish an optimal ruminal microbial community for improved digestion of this low-quality forage to lessen ruminal digesta fill and increase intake [32,33]. In the present experiment, wheat straw DM intake was 0.29, 0.49, 0.61, and 0.66% BW and 23.5, 37.5, 43.2, and 49.9% greater for LS than for HS in periods 2, 3, 4, and 5, respectively. However, this compensation in basal forage intake was incomplete or partial in periods 2 and 3, with total intake in % BW greater for HS vs. LS. Conversely, wheat straw intake was fully compensatory for the difference in supplement intake in % BW in periods 4 and 5, with similar total DM intake for each supplement treatment.

The degree to which high NDF intake restricts DM intake due to ruminal fill varies with factors such as forage type, animal species, BW, and physiological stage. For example, NDF intake limits have been proposed for different animal types, such as 1.1% BW for dairy cattle [29] and 2.1% and 1.76% BW for ewes of 45 and 90 kg BW, respectively [30]. In the present study, it is likely that daily NDF intake capacity (1.63% BW calculated from the intake and NDF concentration in the diet) reached a maximal level for LS sheep in periods 4 and 5, whereas for HS sheep, the intake was probably not limited by NDF ruminal fill (1.22% BW) but rather by other factors such as nutrient and energy absorption in relation to requirements and potential for efficient metabolism [29,34]. This latter finding in the present experiment was similar to one of Lourencon et al. [14] in the first three periods when different levels of supplement were offered, with similar total intake between LS and HS. However, it is important to note the difference in the level of offering of HS between the studies, with 0.8% BW used previously by Lourencon et al. [14] and 1.00% BW in the current experiment. This difference appeared responsible for much of that in total intake between supplement treatments, as the degree to which wheat straw intake was greater for LS than for HS noted by Lourencon et al. [14] was fairly similar to that in the current experiment, with differences of 0.44, 0.44, 0.57, and 0.33% BW in periods 1, 2, 3, and 4, respectively. Overall, based on the total DM intake in these two experiments, it would not seem that there were marked differences in negative associative effects of the HS supplement treatment (i.e., 1.00 vs. 0.80% BW DM intake), with greater benefits in terms of nutrient and energy intake from higher intakes of the HS supplement in the present experiment.

### 4.2. BW and ADG

In contrast to the findings of Lourencon et al. [14] and other studies [11,13,35,36], there were only two measurement days (on days 49 and 225) when differences in BW among breeds were detected, although the values in all periods were numerically lowest for STC. This in part could relate to the limited number of observations and, relatedly, the inclusion of both primiparous and multiparous animals. Because of relatively high variability in BW, even though ADG was greater for HS vs. LS in periods 1, 2, and 3, the differences in BW between the supplement treatments did not occur until day 73. Smaller magnitudes of difference in BW on days 198 and 225 than earlier would relate to imposing the HS treatment on all animals in periods 6 and 7, resulting in greater ADG for LS vs. HS in period 6 and no difference in period 7. That is, higher ADG in period 6 was likely due to compensatory metabolism and tissue accretion as a result of limited nutritional supply in the previous periods [37,38]. Greater ADG during the higher nutritional plane of previously restricted animals has been attributed to factors such as increased feed intake and efficiency of metabolism in the early segment of the realimentation phase as a function of a reduced maintenance requirement because of the decreased mass of metabolically active organs (e.g., digestive tract and liver) in association with endocrine (e.g., insulin and insulin-like growth factor 1) and metabolic (higher protein sparing) responses [37,39,40]. In a study with beef cattle steers, the efficiency of metabolizable energy utilization after a high nutritional plane was elevated for at least 28 days and thereafter decreased steadily to control levels [41]. In period 7, this compensatory response perhaps did not exist, which could also involve the advanced stage of pregnancy. Nonetheless, differences in BW at the end of periods 6 and 7 were substantial (i.e., approximately 21% for HS vs. LS).

### 4.3. BCS and BMI

The differences in BCS (KAT > DOR and STC) at most times despite similar BW suggest considerable disparities in the sites of tissues varying in composition such as lipid. Assuming that BCS is markedly influenced by the presence of subcutaneous fat as well as muscular lean tissue, it would appear that KAT stored relatively more energy in external than internal sites compared with DOR and STC. Breed differences for subcutaneous, carcass, and internal fat deposition that are greatly genetically influenced have been reported in various studies [42,43,44]. However, based on the change in BCS during the experiment, it seems that this was due primarily to deposition and maintenance in earlier periods of time. However, these results are considerably different than noted by Lourencon et al. [14], with no differences among breeds noted at the beginning of the experiment and after 4 wk and then lower BCS for STC than for DOR and KAT after 8 wk. Similarly, the whole body concentrations of water, protein, fat, and energy in that study either were similar among breeds or differed between STC and the other breeds of DOR and KAT. The factors responsible for these differences are unclear, but differences in BW, with higher values for Lourencon et al. [14] vs. the present study, might be one of the factors.

The differences in BMI among breeds are quite dissimilar to those in BCS, which again may reflect varying sites and levels of adipose and lean tissues relative to BW and body size. Breed differences in BMI have been consistent because BMIs are functions of body size (e.g., height, length, circumference), which generally vary among breeds of a species at a particular age [14,45,46]. Overall, similar BMI for DOR and KAT in most instances, in contrast to greater BCS for KAT, suggests a greater mass of internal adipose tissue stores for DOR. Conversely, the lowest BMI for STC among breeds in most cases, despite similar BCS for STC and DOR, implies minimal tissue energy stores both in subcutaneous and internal sites. However, as suggested earlier for BCS, it would appear that such differences developed and were in place before this study.

### 4.4. HR and HE

Heat energy is associated with basal metabolism, activity, and tissue accretion. Heat energy was 1.234 and 2.430 MJ/day and 15.2 and 21.9% greater for HS than for LS on days 49 and 73, respectively. Greater straw intake by LS sheep could have contributed to higher HE associated with mastication during eating and rumination, but it seems that differences in other conditions such as basal metabolism and tissue deposition had a relatively greater impact. Again, high planes of nutrition elicit increased heat production, at least in part because of the greater mass of metabolically active internal organs [47,48,49]. These differences in HE were of considerably lower magnitude than those in total DM intake. Although digestibility and metabolizability, which could be higher in HS vs. LS, may have been affected by supplement treatment, based on these findings, the recovered energy was probably much greater for HS than for LS, in accordance with differences in reproductive performance noted later.

### 4.5. Blood Constituent Concentrations

There were only two interactions between breed and day of sampling, with causal factors unclear. Similarly, interactions between supplement treatment and day of sampling for all variables exemplify the importance of considering the time during which the treatments were implemented on potential differences among nutritional plane treatments.

The tendency for a higher total protein concentration for HS than for LS may relate to greater protein intake for the HS supplement treatment. Similarly, the greater blood urea N concentration in HS vs. LS sheep probably resulted from higher protein intake for HS, consequently leading to greater ruminal ammonia N production by ruminal microbes and, thus, elevated ammonia N absorption into the bloodstream [50,51]. The magnitudes of difference in the total protein and albumin levels for HS sheep were similar throughout periods of the study, but they decreased with advancing time for the LS supplement treatment. This reflects that protein intake for the LS supplement treatment was less than required or that could be efficiently utilized, which was also reflected in ADG as noted before. This is supported by urea N concentrations for LS that were similar among days 14, 49, 73, and 162 and less than on day −7, which contrasts levels for HS greater than for LS on days 14, 49, 73, and 162, though with levels for HS declining from day 49 to 73 and 162.

Blood glucose and NEFA concentrations are indicators of energy status in animals. A greater glucose concentration for HS vs. LS at one time and corresponding numerical differences at all others involve conditions such as greater ruminal microbial propionate production because of corn included in the HS supplement [52], but relatively high variability reflects the influence of many other factors. The concentration of blood NEFA increases when energy demands cannot be met from feed intake and lipolysis occurs to supply precursor molecules for energy [53], which is in accordance with the substantial differences between supplement treatments at all times except day −7. The greater blood cholesterol levels on days 14 and 49 for LS vs. HS further depict the energy deficit of LS sheep, as cholesterol is involved in the transport of fatty acids from adipose tissue in response to an energy shortfall in absorbed fuels [54,55]. Opposite differences between supplement treatments in blood cholesterol concentration on days 73 and 162 were somewhat unexpected given differences at earlier times but may reflect the compensatory increased level of wheat straw intake for LS in this latter segment of the supplementation period to lessen maternal tissue mobilization. The greater TG concentration in HS sheep at all times except day −7 reflects an elevated energy status compared with LS sheep. Factors responsible for the greatest level of TG for HS among days on day 163 and the greatest magnitudes of difference between supplement treatments at this time are unclear since total DM intake relative to BW was similar between periods 4 and 5. However, it is possible that a cumulative effect of differences in energy and nutrients absorbed between supplement treatments was somehow involved.

### 4.6. Reproductive Performance

A high plane of nutrition before breeding has been shown to improve the reproductive performance of sheep in many studies [5,56] via improved reproductive hormonal balance and follicular development [3,57]. Supplement treatment had a much greater effect on reproductive performance measures than in the previous experiment of Lourencon et al. [14], which could involve factors such as the higher level of intake of the HS supplement and a longer length of time during early gestation when supplement treatments were imposed in the current experiment. In the previous experiment, there were no significant effects of supplement treatment on these variables. Conversely, in the current study, the tendency for a difference in the birth rate was substantial, with 78.0 and 96.3% for LS and HS, respectively, as was also the case for litter sizes of 1.37 and 1.68 lambs for LS and HS, respectively. The latter difference resulted in a litter birth weight of 1.08 kg and 19.3% greater for HS vs. LS. It has been suggested that BCS during breeding should range from 3 to 3.5 for optimal reproductive performance [5,58]. The BCS for both HS and LS treatments were in this range. Therefore, it would appear that the difference in reproductive performance was a function of nutritional status during breeding and gestation, which are important for the embryo quality, ovulation rate, uterine response, and reproductive performance of ewes [4]. However, differences in experimental conditions between the present study and that of Lourencon et al. [14] could have had an influence. For example, the trial of Lourencon et al. [14] was with mostly multiparous animals (i.e., 81 of 85), with an average initial age of all females of 4.9 years. In contrast, the current experiment entailed a greater number of relatively young ewe lambs (72; 1.5 years initial age) that had not previously given birth compared with multiparous ewes (35; 5.6 years initial age). A greater number of multiparous ewes would be desirable for a stronger evaluation of the potential effects of parity, although the results of the separate analysis conducted suggest fairly similar effects of supplement treatments.

### 4.7. BMI and BCS Relationships

The correlation coefficients between BMI–WH and BMI–WP and BW were at least slightly greater than those for BMI–GH and BMI–GP, suggesting less variability in measurement or among animals in height than heart girth. Relatedly, higher correlations between change in BMI–WH and BMI–WP and that in BCS from day −7 to 162 relative to BMI–GH and BMI–GP also implies advantages from the use of BMI–WH and BMI–WP. Likewise, as observed in studies such as Lourencon et al. [14] and Wang et al. [19], the BMI measurements in this study, particularly BMI–WH and BMI–WP, are more highly related to BW than BCS, although, naturally, it should be realized that BMI is calculated from BW and may not be as reflective of sites of different types of tissues relative to BCS.

## 5. Summary and Conclusions

The supplement treatments imposed before breeding and in early gestation caused differences in wheat straw and total feed intake, daily body weight gain, body condition score and mass indexes, litter size, total litter birth weight, and heat energy production, although the effects were generally similar among breeds. Likewise, breed affected many factors such as body weight, condition score, and mass indexes, but reproductive performance was not influenced. The litter size and birth weight for the supplement treatments reflect the importance of considering not only the need for additional protein during breeding and early gestation periods with the consumption of a low-quality basal dietary forage but also the inclusion of a high-energy feedstuff(s) for higher reproductive performance.

## Figures and Tables

**Figure 1 animals-13-00814-f001:**
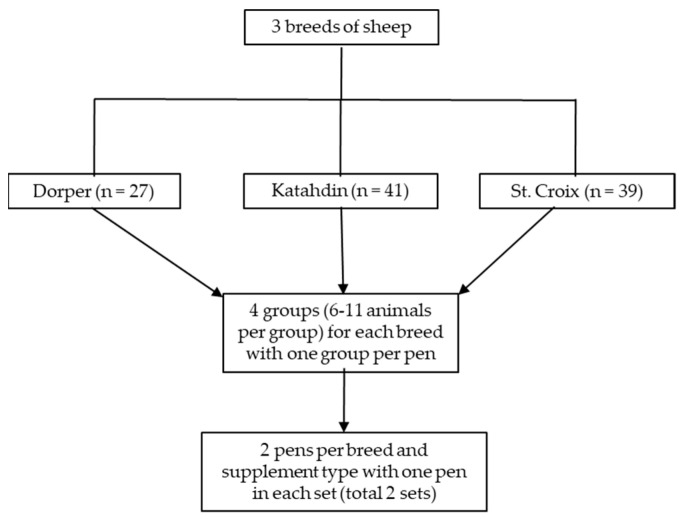
Schematic presentation of the experimental design.

**Figure 2 animals-13-00814-f002:**
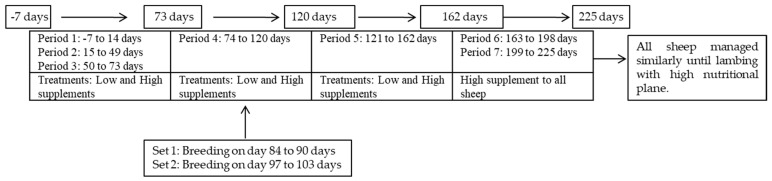
Two supplements fed to three breeds of sheep during different periods.

**Table 1 animals-13-00814-t001:** Composition of wheat straw and supplements (%, dry matter basis) ^1^.

	Wheat Straw		Low Supplement		High Supplement	
Item	Mean	SEM	Mean	SEM	Mean	SEM
Ash	6.4	0.23	8.6	0.12	3.6	0.10
Crude protein	3.9	0.11	47.8	0.19	14.5	0.67
Neutral detergent fiber	80.2	0.52	12.1	0.55	13.5	0.38
Acid detergent fiber	53.9	0.38	8.8	0.36	5.3	0.24
Acid detergent lignin	10.4	0.20				

^1^ Low supplement = soybean meal; High supplement = 25% soybean meal and 75% rolled corn; mean of weekly samples.

**Table 2 animals-13-00814-t002:** Effects of breed and supplement treatment on dry matter intake by hair sheep ^1^.

	Effect *p* Value ^2^			Dorper		Katahdin		St. Croix			
Item	Brd	Sup	Brd × Sup	Low	High	Low	High	Low	High	SEM	Brd ^3^
Period 1											
g/day											
Supplement	0.266	<0.001	0.332	87	556	83	579	79	494	25.6	
Wheat straw	0.648	0.337	0.907	640	618	626	550	679	616	62.8	
Total	0.906	<0.001	0.789	727	1174	709	1296	758	1110	70.4	
% body weight											
Supplement	0.588	<0.001	0.290	0.15	0.99	0.14	0.99	0.15	0.98	0.007	
Wheat straw	0.211	0.461	0.916	1.13	1.10	1.07	0.94	1.32	1.22	0.131	
Total	0.208	<0.001	0.918	1.28	2.09	1.21	1.93	1.47	2.20	0.131	
Period 2											
g/day											
Supplement	0.266	<0.001	0.332	87	556	83	579	79	494	25.6	
Wheat straw	0.410	0.045	0.927	920	750	801	681	830	700	67.9	
Total	0.407	0.002	0.818	1007	1306	884	1261	909	1193	77.0	
% body weight											
Supplement	0.636	<0.001	0.939	0.16	0.96	0.14	0.95	0.16	0.95	0.013	
Wheat straw	0.233	0.036	0.951	1.65	1.29	1.39	1.12	1.64	1.35	0.139	
Total	0.214	0.005	0.954	1.80	2.25	1.53	2.06	1.80	2.30	0.140	
Period 3											
g/day											
Supplement	0.266	<0.001	0.332	87	556	83	579	79	494	25.6	
Wheat straw	0.410	0.006	0.338	1031	752	879	773	902	739	54.6	
Total	0.312	0.002	0.329	1117	1308	962	1353	981	1233	62.6	
% body weight											
Supplement	0.063	<0.001	0.076	0.16	0.94	0.15	0.92	0.16	0.93	0.009	
Wheat straw	0.321	0.005	0.638	1.89	1.27	1.59	1.24	1.88	1.39	0.135	
Total	0.292	0.039	0.648	2.04	2.21	1.74	2.17	2.04	2.32	0.135	
Period 4											
g/day											
Supplement	0.266	<0.001	0.332	87	556	83	579	79	494	25.6	
Wheat straw	0.061	0.002	0.507	1111	870	990	800	938	801	42.1	
Total	0.040	<0.001	0.711	1198	1426	1073	1379	1017	1295	46.2	S < D
% body weight											
Supplement	0.063	<0.001	0.117	0.17	0.95	0.16	0.91	0.17	0.91	0.009	
Wheat straw	0.253	0.001	0.986	2.12	1.49	1.89	1.27	2.07	1.48	0.130	
Total	0.213	0.227	0.988	2.29	2.45	2.05	2.18	2.24	2.39	0.133	
Period 5											
g/day											
Supplement	0.281	<0.001	0.315	87	556	83	579	81	494	25.6	
Wheat straw	0.441	0.001	0.365	989	814	974	795	910	813	29.5	
Total	0.298	<0.001	0.963	1076	1370	1056	1375	992	1306	47.2	
% body weight											
Supplement	0.063	<0.001	0.117	0.17	0.93	0.16	0.86	0.18	0.88	0.011	
Wheat straw	0.115	<0.001	0.789	1.96	1.35	1.91	1.18	2.08	1.45	0.088	
Total	0.088	0.425	0.643	2.13	2.28	2.07	2.04	2.27	2.33	0.091	
Average											
g/day											
Supplement	0.269	<0.001	0.328	87	556	83	579	80	494	25.6	
Wheat straw	0.350	0.007	0.789	938	761	854	720	852	734	43.7	
Total	0.306	<0.001	0.771	1025	1317	937	1299	931	1227	53.9	
% body weight											
Supplement	0.020	<0.001	0.058	0.16	0.96	0.15	0.92	0.17	0.93	0.005	K < D
Wheat straw	0.191	0.003	0.992	1.75	1.30	1.57	1.15	1.80	1.38	0.112	
Total	0.165	0.009	0.999	1.91	2.26	1.72	2.07	1.96	2.31	0.113	

^1^ Supplement treatments were soybean meal fed at approximately 0.15% of initial body weight (determined 7 days before the start of the feeding period) and a mixture of 25% soybean meal and 75% ground corn given at approximately 1% of initial body weight (dry matter basis; Low and High, respectively); periods 1, 2, 3, 4, and 5 were 14, 35, 24, 47, and 42 days in length, respectively; with estrus synchronization, a 7-day breeding period began for set 1 animals (one of the two pens per treatment) after 83 days of feeding on day 12 of period 4 and for set 2 animals after 96 days of feeding on day 24 of period 4. ^2^ Brd = breed; Sup = supplement treatment. ^3^ D = Dorper; K = Katahdin; S = St. Croix; main effect mean differences (*p* < 0.05) with nonsignificant interactions between breed and supplement treatment (*p* > 0.05).

**Table 3 animals-13-00814-t003:** Effects of breed and supplement treatment on body weight and average daily gain of hair sheep ^1^.

	Effect *p*-Value ^2^			Dorper		Katahdin		St. Croix			
Item ^3^	Brd	Sup	Brd × Sup	Low	High	Low	High	Low	High	SEM	Brd ^4^
BW (kg), day											
−7	0.106	0.583	0.969	57.7	55.6	58.5	57.9	51.3	49.4	3.31	
14	0.082	0.865	0.974	56.4	56.6	58.2	59.3	51.7	51.6	2.71	
49	0.046	0.144	0.838	55.7	59.4	57.1	63.1	49.7	52.3	2.98	S < K
73	0.128	0.039	0.869	53.8	58.8	53.4	62.0	46.7	54.2	3.25	
120	0.140	0.020	0.625	51.1	57.6	51.3	65.9	44.2	54.1	3.38	
162	0.092	0.002	0.730	50.1	62.7	51.2	69.7	43.5	57.9	3.69	
198	0.088	0.004	0.688	55.9	67.3	53.8	67.6	49.8	58.4	3.03	
225	0.029	0.004	0.681	63.9	73.7	60.1	75.5	52.6	63.6	3.27	S < D & K
ADG (g), period											
1	0.279	0.022	0.901	−62	51	−16	64	18	105	37.1	
2	0.039	0.001	0.273	−22	76	−32	110	−59	19	19.3	S < K
3	0.247	0.014	0.321	−79	−22	−154	−46	−123	79	43.7	
4	0.250	0.046	0.373	−57	−29	−46	83	−56	−1	34.6	
5	0.957	0.006	0.739	−24	120	−1	90	−14	90	33.3	
6	0.022	0.017	0.258	130	113	48	−59	153	12	33.1	K < D & S
7	0.067	0.501	0.307	295	238	232	273	102	195	43.8	
1 to 5	0.175	<0.001	0.256	−46	42	−44	70	−47	51	7.3	
6 to 7	0.086	0.172	0.990	200	166	126	84	131	90	30.8	
1 to 7	0.221	<0.001	0.770	20	76	6	73	2	62	7.7	

^1^ Supplement treatments were soybean meal fed at approximately 0.15% of initial body weight (determined 7 days before the start of the feeding period) and a mixture of 25% soybean meal and 75% ground corn given at approximately 1% of initial body weight (dry matter basis; Low and High, respectively). ^2^ Brd = breed; Sup = supplement treatment. ^3^ BW = body weight; ADG = average daily gain; BW was determined 7 days before the supplementation treatments were imposed (day −7), after 14, 49, 73, 120, and 162 days of feeding, and after additional periods of 36 and 27 days (days 198 and 225, respectively); periods 1, 2, 3, 4, 5, 6, and 7 were 21 (7 days before supplementation began and 14 days thereafter), 35, 24, 47, 42, 36, and 27 days in length, respectively. ^4^ D = Dorper; K = Katahdin; S = St. Croix; main effect mean differences (*p* < 0.05) with nonsignificant interactions between breed and supplement treatment (*p* > 0.05).

**Table 4 animals-13-00814-t004:** Effects of breed and supplement treatment on body condition score of hair sheep ^1^.

	Effect *p* Value ^2^			Dorper		Katahdin		St. Croix			
Item ^3^	Brd	Sup	Brd × Sup	Low	High	Low	High	Low	High	SEM	Brd ^4^
Day											
−7	0.022	0.014	0.678	3.20	2.90	3.43	3.25	3.16	2.81	0.098	K > D & S
14	0.108	0.398	0.908	3.14	3.11	3.36	3.26	3.16	3.09	0.085	
49	0.003	0.031	0.703	2.97	3.15	3.23	3.31	2.89	3.02	0.056	K > D & S
73	0.019	0.018	0.579	3.11	3.27	3.26	3.56	2.90	3.21	0.092	K > D & S
120	0.006	0.001	0.852	2.76	3.19	2.98	3.49	2.55	3.05	0.086	K > D & S
162	0.028	<0.001	0.286	2.59	3.25	2.89	3.52	2.77	3.16	0.083	K > D & S
198	0.007	<0.001	0.598	3.01	3.41	2.67	3.15	2.78	3.15	0.059	D > K & S
225	0.008	<0.001	0.012	3.04 ^bc^	3.23 ^cd^	2.73 ^a^	3.37 ^d^	2.74 ^a^	2.98 ^b^	0.057	
Change, day											
−7 to 14	0.159	0.015	0.375	−0.07	0.22	−0.07	0.01	0.00	0.28	0.077	
14 to 49	0.061	0.003	0.962	−0.16	0.03	−0.13	0.05	−0.26	−0.07	0.046	
49 to 73	0.749	0.049	0.126	0.14	0.09	0.03	0.25	0.01	0.19	0.058	
73 to 120	0.481	0.004	0.725	−0.35	−0.05	−0.29	−0.06	−0.35	−0.16	0.065	
120 to 162	0.007	0.082	0.029	−0.17 ^a^	0.07 ^bc^	−0.08 ^ab^	0.03 ^ab^	0.22 ^c^	0.11 ^bc^	0.047	
162 to 198	<0.001	0.151	0.345	0.30	0.15	−0.25	−0.37	−0.04	−0.01	0.059	K < S < D
198 to 225	0.003	0.115	0.021	0.02 ^b^	−0.18 ^a^	0.07 ^bc^	0.20 ^c^	−0.03 ^ab^	−0.17 ^a^	0.047	
−7 to 162	0.334	<0.001	0.384	−0.61	0.36	−0.53	0.27	−0.39	0.34	0.077	
162 to 225	0.003	0.020	0.049	0.33 ^b^	−0.03 ^a^	−0.19 ^a^	−0.16 ^a^	−0.07 ^a^	−0.18 ^a^	0.058	
−7 to 225	0.059	<0.001	0.481	−0.28	0.31	−0.71	0.10	−0.42	0.18	0.102	

^1^ Supplement treatments were soybean meal fed at approximately 0.15% of initial BW (determined 7 days before the start of the feeding period) and a mixture of 25% soybean meal and 75% ground corn given at approximately 1% of initial BW (DM basis; Low and High, respectively). ^2^ Brd = breed; Sup = supplement treatment. ^3^ Body condition score (1–5) was determined 7 days before the supplementation treatments were imposed (day −7), after 14, 49, 73, 120, and 162 days of feeding, and after additional periods of 36 and 27 days (days 198 and 225, respectively); periods 1, 2, 3, 4, 5, 6, and 7 were 21 (7 days before supplementation began and 14 days thereafter), 35, 24, 47, 42, 36, and 27 days in length, respectively. ^4^ D = Dorper; K = Katahdin; S = St. Croix; main effect mean differences (*p* < 0.05) with nonsignificant interactions between breed and supplement treatment (*p* > 0.05). ^a–d^ Means without a common superscript letter within a variable differ (*p* < 0.05).

**Table 5 animals-13-00814-t005:** Effects of breed and supplement treatment on body mass indexes of hair sheep ^1^.

	Effect *p* Value ^2^			Dorper		Katahdin		St. Croix			
Item ^3^	Brd	Sup	Brd × Sup	Low	High	Low	High	Low	High	SEM	Brd ^4^
Initial, day −7											
BMI–WH	0.023	0.396	0.809	14.30	14.33	14.73	14.16	13.06	12.49	0.495	S < D & K
BMI–WP	0.044	0.293	0.986	11.62	11.29	11.63	11.16	10.40	9.94	0.443	S < D & K
BMI–GH	0.034	0.343	0.695	10.10	10.16	10.62	10.26	9.74	9.34	0.271	S < K
BMI–GP	0.069	0.219	0.968	8.20	8.01	8.38	8.09	7.76	7.44	0.234	
Day 14											
BMI–WH	0.015	0.763	0.643	14.02	13.64	13.55	13.83	12.04	12.46	0.414	S < D & K
BMI–WP	0.010	0.964	0.648	11.12	10.80	10.61	10.66	9.42	9.72	0.307	S < D & K
BMI–GH	0.031	0.764	0.453	10.42	10.12	10.30	10.56	9.60	9.82	0.218	S < K
BMI–GP	0.035	0.943	0.561	8.26	8.01	8.06	8.14	7.52	7.67	0.176	S < D & K
Day 49											
BMI–WH	0.004	0.087	0.721	13.85	14.16	13.26	14.31	11.51	12.29	0.426	S < D & K
BMI–WP	0.005	0.157	0.810	10.99	11.18	10.40	11.04	9.13	9.61	0.330	S < D & K
BMI–GH	0.011	0.436	0.565	10.63	10.51	10.32	10.81	9.43	9.62	0.266	S < D & K
BMI–GP	0.021	0.802	0.734	8.43	8.30	8.10	8.35	7.48	7.52	0.234	S < D & K
Day 73											
BMI–WH	0.057	0.039	0.940	13.33	14.70	12.60	13.92	11.01	12.77	0.690	
BMI–WP	0.092	0.048	0.922	10.34	11.27	9.81	10.74	8.76	10.05	0.517	
BMI–GH	0.064	0.083	0.752	10.07	10.64	10.02	10.43	8.81	9.79	0.384	
BMI–GP	0.117	0.115	0.677	7.82	8.15	7.82	8.04	7.02	7.71	0.277	
Day 162											
BMI–WH	0.034	0.001	0.748	10.94	14.54	12.06	15.09	10.02	12.57	0.663	S < K
BMI–WP	0.004	<0.001	0.694	9.63	11.35	9.44	11.71	8.04	10.11	0.305	S < D & K
BMI–GH	0.079	0.015	0.810	8.52	10.14	9.34	10.53	8.14	9.16	0.451	
BMI–GP	0.006	0.003	0.442	7.50	7.92	7.31	8.18	6.53	7.38	0.176	S < D & K
Change, day −7 to 14											
BMI–WH	0.647	0.065	0.088	−0.28	−0.70	−1.18	−0.33	−1.14	−0.02	0.280	
BMI–WP	0.624	0.077	0.356	−0.50	−0.49	−1.02	−0.51	−1.07	−0.22	0.262	
BMI–GH	0.672	0.096	0.074	0.33	−0.04	−0.32	0.30	−0.21	0.48	0.192	
BMI–GP	0.599	0.086	0.294	0.066	0.01	−0.31	0.04	−0.29	0.22	0.161	
Change, day 14 to 49											
BMI–WH	0.155	0.026	0.682	−0.17	0.53	−0.29	0.48	−0.53	−0.17	0.252	
BMI–WP	0.217	0.029	0.504	−0.13	0.38	−0.21	0.38	−0.29	−0.11	0.181	
BMI–GH	0.111	0.446	0.759	0.21	0.39	0.02	0.25	−0.17	−0.20	0.191	
BMI–GP	0.149	0.601	0.580	0.17	0.28	0.03	0.21	−0.05	−0.14	0.140	
Change, day 49 to 73											
BMI–WH	0.358	0.052	0.592	−0.52	0.52	−0.68	−0.38	−0.51	0.47	0.392	
BMI–WP	0.241	0.033	0.624	−0.65	0.07	−0.61	−0.30	−0.37	0.44	0.270	
BMI–GH	0.836	0.051	0.209	−0.57	0.12	−0.32	−0.38	−0.63	0.17	0.237	
BMI–GP	0.407	0.066	0.320	−0.009	0.004	−0.005	−0.005	−0.007	0.003	0.0039	
Change, day 73 to 162											
BMI–WH	0.048	0.007	0.390	−2.38	−0.16	−0.55	1.17	−0.99	−0.19	0.486	D < K
BMI–WP	0.051	0.002	0.448	−0.71	0.11	−0.36	0.97	−0.72	0.06	0.233	
BMI–GH	0.151	0.053	0.335	−1.55	−0.50	−0.68	0.10	−0.68	−0.63	0.318	
BMI–GP	0.356	0.063	0.223	−0.32	−0.22	−0.50	0.13	−0.48	−0.33	0.176	
Change, day −7 to 162											
BMI–WH	0.462	<0.001	0.910	−3.36	0.20	−2.70	0.93	−3.08	0.09	0.579	
BMI–WP	0.629	<0.001	0.546	−1.99	0.07	−2.19	−0.55	−2.39	0.17	0.297	
BMI–GH	0.709	0.009	0.991	−1.58	−0.03	−1.28	0.27	−1.62	−0.18	0.493	
BMI–GP	0.446	0.001	0.345	−0.70	−0.09	−1.06	0.08	−1.24	−0.07	0.189	

^1^ Supplement treatments were soybean meal fed at approximately 0.15% of initial body weight (determined 7 days before the start of the feeding period) and a mixture of 25% soybean meal and 75% ground corn given at approximately 1% of initial body weight (dry matter basis; Low and High, respectively). ^2^ Brd = breed; Sup = supplement treatment. ^3^ Body mass indexes (BMI) were determined 7 days before the supplementation treatments were imposed (day −7) and after 14, 49, 73, and 162 days of feeding; Wither = height at withers; Hook = point of the shoulder to hook bone; Pin = point of the shoulder to pin bone; Heart = heart girth; BMI–WH = BW/(Wither × Hook) [g/cm^2^]; BMI–WP = BW/(Wither × Pin) [g/cm^2^]; BMI–GH = BW/(Heart × Hook) [g/cm^2^]; BMI–GP = BW/(Heart × Pin) [g/cm^2^]. ^4^ D = Dorper; K = Katahdin; S = St. Croix; main effect mean differences (*p* < 0.05) with nonsignificant interactions between breed and supplement treatment (*p* > 0.05).

**Table 6 animals-13-00814-t006:** Effects of breed and supplement treatment on heart rate and heat energy by hair sheep ^1^.

	Effect *p* Value ^2^			Dorper		Katahdin		St. Croix		
Item ^3^	Brd	Sup	Brd × Sup	Low	High	Low	High	Low	High	SEM
HE:HR (kJ/kg BW^0.75^/beat)	0.493	0.778	0.886	6.39	6.16	6.79	6.65	6.52	6.63	0.352
Day 14										
HR (beats/min)	0.786	0.043	0.722	72.2	85.8	69.9	85.9	71.5	78.6	5.85
HE (kJ/kg BW^0.75^)	0.995	0.219	0.959	468	523	458	545	468	526	59.4
HE (MJ/day)	0.729	0.276	0.885	9.87	10.74	9.51	11.65	9.02	10.03	1.610
Day 49										
HR (beats/min)	0.282	0.005	0.326	66.8	92.1	63.9	78.4	68.3	77.8	4.67
HE (kJ/kg BW^0.75^)	0.858	0.023	0.647	423	569	428	519	445	511	40.8
HE (MJ/day)	0.220	0.004	0.302	63.0	84.4	65.3	79.5	81.2	82.6	5.83
Day 73										
HR (beats/min)	0.272	0.041	0.302	63.0	84.4	65.3	79.5	81.2	82.6	5.83
HE (kJ/kg BW^0.75^)	0.287	0.120	0.650	394	510	439	524	528	549	49.8
HE (MJ/day)	0.763	0.039	0.803	7.94	10.83	8.67	11.51	9.39	10.95	1.127

^1^ Supplement treatments were soybean meal fed at approximately 0.15% of initial body weight (determined 7 days before the start of the feeding period) and a mixture of 25% soybean meal and 75% ground corn given at approximately 1% of initial body weight (dry matter basis; Low and High, respectively). ^2^ Brd = breed; Sup = supplement treatment. ^3^ HE = heat energy; HR = heart rate; HR was measured on one set of animals 7 days before days 14, 49, and 73 and 7 days after, with HE based on HR and HE:HR determined after the 162-day supplementation period.

**Table 7 animals-13-00814-t007:** *p*-values for effects of breed, supplement treatment, and day on blood constituent concentrations in hair sheep ^1^.

	Source of Variation ^2^						
Item	Brd	Sup	Brd × Sup	Day	Brd × Day	Sup × Day	Brd × Sup × Day
Total protein (g/dL)	0.009	0.056	0.351	<0.001	0.516	0.012	0.585
Albumin (g/dL)	0.213	0.396	0.665	0.046	0.433	<0.001	0.822
Urea nitrogen (mg/dL)	0.034	<0.001	0.383	<0.001	0.106	<0.001	0.327
Cholesterol (mg/dL)	0.478	0.441	0.934	<0.001	0.569	<0.001	0.872
Triglycerides (mg/dL)	0.004	<0.001	0.021	<0.001	0.323	<0.001	0.761
Glucose (mg/dL)	0.610	0.475	0.644	<0.001	<0.001	0.008	0.056
Lactate (mg/dL)	0.282	0.129	0.205	0.016	0.036	0.011	0.349
Nonesterified fatty acids (mEq/L)	0.002	<0.001	0.072	<0.001	0.800	<0.001	0.330
Total antioxidant activity (µM)	0.350	0.270	0.743	<0.001	0.035	0.007	0.695

^1^ Supplement treatments were soybean meal fed at approximately 0.15% of initial body weight (determined 7 days before the start of the feeding period) and a mixture of 25% soybean meal and 75% ground corn given at approximately 1% of initial body weight (dry matter basis; Low and High, respectively). ^2^ Brd = breed (Dorper, Katahdin, and St. Croix); Sup = supplement treatment; Day = 7 days before the supplementation treatments were imposed (day −7) and after 14, 49, 73, and 162 days of feeding.

**Table 8 animals-13-00814-t008:** Effects of breed, supplementation treatment, and period on blood constituent concentrations in hair sheep ^1^.

	Interaction ^2^		Breed ^3^				Sup			Day ^4^					
Item ^5^	Brd	Sup	DOR	KAT	STC	SEM	Low	High	SEM	−7	14	49	73	162	SEM
TP (g/dL)			6.66 ^a^	6.95 ^b^	6.99 ^b^	0.052									
		Low								7.09 ^e^	6.92 ^c–e^	6.91 ^c–e^	6.64 ^b^	6.41 ^a^	0.074
		High								7.01 ^de^	7.02 ^de^	7.01 ^de^	6.86 ^bc^	6.79 ^bc^	
ALB (g/dL)			2.66	2.74	2.66	0.034									
		Low								2.79 ^c^	2.78 ^c^	2.79 ^c^	2.63 ^b^	2.54 ^a^	0.039
		High								2.64 ^b^	2.64 ^b^	2.63 ^b^	2.71 ^bc^	2.73 ^bc^	
UN (mg/dL)			18.4 ^ab^	17.6 ^a^	19.3 ^b^	0.36									
		Low								20.7 ^c^	15.6 ^a^	15.7 ^a^	15.3 ^a^	15.4 ^a^	0.52
		High								20.2 ^c^	22.4 ^d^	22.5 ^d^	19.3 ^c^	17.1 ^b^	
TG (mg/dL)										26.5 ^a^	27.5 ^a^	27.6 ^a^	28.8 ^a^	34.9 ^b^	0.89
		Low	25.3 ^a^	26.5 ^a^	23.9 ^a^	1.06				26.4 ^a^	24.2 ^a^	24.3 ^a^	25.2 ^a^	26.2 ^a^	5.94
		High	37.6 ^c^	33.7 ^b^	27.4 ^a^					26.6 ^a^	30.8 ^b^	30.9 ^b^	32.4 ^b^	43.7 ^c^	
CHOL (mg/dL)			71.3	67.3	68.8	2.89	70.1	68.1	1.76						
		Low								71.5 ^c^	80.0 ^d^	80.1 ^d^	62.0 ^b^	57.1 ^a^	2.20
		High								70.8 ^c^	64.6 ^b^	64.5 ^b^	70.5 ^c^	69.9 ^c^	
Glucose (mg/dL)			55.8	48.4	57.6	6.89	50.9	57.0	5.63						
		Low								62.3 ^cd^	48.7 ^b^	44.3 ^a^	51.0 ^b^	48.4 ^b^	5.69
		High								65.0 ^d^	55.4 ^bc^	53.4 ^bc^	55.5 ^bc^	55.6 ^bc^	
	DOR									69.9 ^f^	53.9 ^d–f^	49.3 ^bc^	53.8 ^d–f^	52.2 ^b–e^	6.97
	KAT									53.3 ^c–f^	47.0 ^ab^	45.7 ^a^	49.1 ^b^	46.8 ^ab^	
	STC									67.7 ^ef^	55.2 ^ef^	51.6 ^b–d^	56.8 ^ef^	56.9 ^ef^	
Lactate (mg/dL)															
		Low								18.5 ^cd^	10.9 ^a^	11.3 ^a^	14.3 ^ab^	14.0 ^ab^	1.69
		High								15.2 ^a–d^	16.4 ^b–d^	14.9 ^a–c^	17.8 ^b–d^	18.8 ^d^	
	DOR									21.5 ^e^	14.9 ^a–d^	14.4 ^a–c^	15.2 ^a–d^	20.3 ^de^	2.06
	KAT									11.8 ^ab^	13.1 ^a–c^	13.7 ^a–c^	16.5 ^a–e^	14.5 ^a–d^	
	STC									17.3 ^c–e^	12.9 ^ab^	11.3 ^a^	16.6 ^b–e^	14.4 ^a–d^	
NEFA (mEq/L)			0.244 ^a^	0.374 ^b^	0.354 ^b^	0.0185									
		Low								0.312 ^c^	0.656 ^e^	0.469 ^d^	0.418 ^d^	0.407 ^d^	0.0266
		High								0.293 ^c^	0.171 ^ab^	0.132 ^a^	0.173 ^ab^	0.207 ^b^	
TAC (µM)			251	253	264	6.3	260	252	5.2						
		Low								264 ^bc^	240 ^ab^	283 ^c^	281 ^c^	234 ^ab^	8.8
		High								260 ^bc^	265 ^bc^	263 ^bc^	246 ^b^	225 ^a^	
	DOR									264 ^cd^	223 ^a^	276 ^cd^	258 ^b–d^	235 ^ab^	11.1
	KAT									244 ^a–c^	268 ^cd^	263 ^cd^	259 ^cd^	231 ^ab^	
	STC									279 ^d^	267 ^cd^	281 ^d^	272 ^cd^	221 ^a^	

^1^ Supplement treatments were soybean meal fed at approximately 0.15% of initial body weight (determined 7 days before the start of the feeding period) and a mixture of 25% soybean meal and 75% ground corn given at approximately 1% of initial body weight (dry matter basis; Low and High, respectively). ^2^ Brd = breed; Sup = supplement treatment. ^3^ D = Dorper; K = Katahdin; S = St. Croix. ^4^ Blood was sampled 7 days before the supplementation treatments were imposed (day −7), after 14, 49, 73, and 162 days of feeding. ^5^ TP = total protein; ALB = albumin; UN = urea nitrogen; CHOL = cholesterol; TG = triglycerides; NEFA = nonesterified fatty acids; TAC = total antioxidant capacity. ^a–f^ Means within grouping without a common superscript letter differ (*p* < 0.05).

**Table 9 animals-13-00814-t009:** Effects of breed and supplement treatment on reproductive performance of hair sheep ^1^.

	Effect *p* Value ^2^			Dorper		Katahdin		St. Croix			
Item ^3^	Brd	Sup	Brd × Sup	Low	High	Low	High	Low	High	SEM	Brd ^4^
Birth rate (%)	0.520	0.063	0.738	66.7	93.5	84.6	95.5	82.8	100.0	9.83	
Litter size	0.190	0.046	0.184	1.38	1.29	1.38	1.95	1.36	1.82	0.152	
Fecundity	0.174	0.021	0.619	0.92	1.21	1.17	1.86	1.12	1.82	0.221	
Birth weight (kg)											
Individual lamb	0.035	0.787	0.329	4.50	4.61	4.38	3.98	3.73	3.88	0.201	S < D
Total litter	0.221	0.049	0.294	5.84	5.74	5.92	7.52	5.04	6.78	0.529	
Gestation length (days)	0.228	0.739	0.661	145.4	144.6	146.2	147.0	144.9	145.7	0.88	

^1^ Supplement treatments were soybean meal fed at approximately 0.15% of initial body weight (determined 7 days before the start of the feeding period) and a mixture of 25% soybean meal and 75% ground corn given at approximately 1% of initial body weight (dry matter basis; Low and High, respectively). ^2^ Brd = breed; Sup = supplement treatment. ^3^ Birth rate and litter size were analyzed with the GLIMMIX procedure and individual lamb and total litter birth weights were analyzed with the MIXED procedure of SAS; gestation length was based on the last day of breeding observed and the date of birth. ^4^ K = Katahdin; S = St. Croix; main effect mean differences (*p* < 0.05) with nonsignificant interactions between breed and supplement treatment (*p* > 0.05).

**Table 10 animals-13-00814-t010:** Correlation coefficients (r) between body mass indexes and body weight and body condition score at different times and change during the supplementation phase of hair sheep ^1^.

			Body Mass Index ^2^				
Day	Item ^3^	Parameter	WH	WP	GH	GP	BCS ^3^
−7	BW	r	0.89	0.91	0.82	0.87	0.42
		*p*	<0.001	<0.001	<0.001	<0.001	<0.001
	BCS	r	0.48	0.50	0.43	0.45	
		*p*	<0.001	<0.001	<0.001	<0.001	
14	BW	r	0.88	0.88	0.83	0.82	0.50
		*p*	<0.001	<0.001	<0.001	<0.001	<0.001
	BCS	r	0.45	0.47	0.41	0.42	
		*p*	<0.001	<0.001	<0.001	<0.001	
49	BW	r	0.88	0.88	0.77	0.75	0.59
		*p*	<0.001	<0.001	<0.001	<0.001	<0.001
	BCS	r	0.58	0.55	0.55	0.52	
		*p*	<0.001	<0.001	<0.001	<0.001	
73	BW	r	0.84	0.84	0.77	0.71	0.62
		*p*	<0.001	<0.001	<0.001	<0.001	<0.001
	BCS	r	0.57	0.54	0.53	0.44	
		*p*	<0.001	<0.001	<0.001	<0.001	
162	BW	r	0.88	0.92	0.78	0.82	0.77
		*p*	<0.001	<0.001	<0.001	<0.001	<0.001
	BCS	r	0.70	0.70	0.58	0.56	
		*p*	<0.001	<0.001	<0.001	<0.001	
−7 to 162 ^3^	BW	r	0.88	0.90	0.74	0.77	0.75
		*p*	<0.001	<0.001	<0.001	<0.001	<0.001
	BCS	r	0.63	0.67	0.45	0.48	
		*p*	<0.001	<0.001	<0.001	<0.001	

^1^ Body mass indexes (BMI) were determined 7 days before the supplementation treatments were imposed (day −7) and after 14, 49, 73, and 162 days of feeding, and body weight (BW) and body condition score (BSC) were determined at these and other times. ^2^ Wither = height at withers; Hook = point of the shoulder to hook bone; Pin = point of the shoulder to pin bone; Heart = heart girth; BMI–WH = BW/(Wither × Hook) [g/cm^2^]; BMI–WP = BW/(Wither × Pin) [g/cm^2^]; BMI–GH = BW/(Heart × Hook) [g/cm^2^]; BMI–GP = BW/(Heart × Pin) [g/cm^2^]. ^3^ Change in BMI, BW, and BCS.

## Data Availability

Data are presented in the tables.

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
