# Peer review of "Effects of the Level and Composition of Concentrate Supplements before Breeding and in Early Gestation on Production of Different Hair Sheep Breeds"

_animals, 2023, doi:10.3390/ani13050814_

Round 1

Reviewer 1 Report

Title: Effects of the level and composition of concentrate supplements before breeding and in early gestation on production of different hair sheep breeds

General comments:

This manuscript determined effects of different levels of intake and composition of supplemental concentrate before breeding and in early gestation on production of three hair sheep consuming a low-quality basal dietary forage. Generally, introduction is sufficient to explain why this study was conducted. Material and method is clearly described. Result is interesting. Figures and Tables are professional.

Specific comments:

Please put three figures including Dorper, Katahdin, and St. Croix hair sheep in Figure 1

Line120-121 Please describe how randomly assigned two supplement treatments to the groups, using a software or using a table of random number.

Line121-124 If possible, please put a figure to show the facility.

Line 166 Pregnancy was diagnosed at 40 days after breeding using ultra-sound?

Line222 Most data were analyzed with the MIXED procedure of SAS, however, which factor is random effect is not clearly.

In 4.6. Reproductive performance section. This study including multiparous and primiparous sheep. It is better to add some discussion to discuss the parity may effect reproductive performance.

Author Response

General comments:

This manuscript determined effects of different levels of intake and composition of supplemental concentrate before breeding and in early gestation on production of three hair sheep consuming a low-quality basal dietary forage. Generally, introduction is sufficient to explain why this study was conducted. Material and method is clearly described. Result is interesting. Figures and Tables are professional.

Response: Thanks for your appreciation of this study and useful suggestions.

Specific comments:

Please put three figures including Dorper, Katahdin, and St. Croix hair sheep in Figure 1

Response: We feel one figure instead of three figures makes same clarity as we have mentioned under box for each breed.

Line120-121 Please describe how randomly assigned two supplement treatments to the groups, using a software or using a table of random number.

Response:  The allocation was performed with a table of random numbers.

Line121-124 If possible, please put a figure to show the facility.

Response:   The description seems adequate.

Line 166 Pregnancy was diagnosed at 40 days after breeding using ultra-sound?

Response: Thanks. It was diagnosed using ultrasonography by ventral external examination with a 3.5 MHz linear-array transducer.

Line222 Most data were analyzed with the MIXED procedure of SAS, however, which factor is random effect is not clearly.

Response: Thanks. We have provided this information now.

In 4.6. Reproductive performance section. This study including multiparous and primiparous sheep. It is better to add some discussion to discuss the parity may effect reproductive performance.

Response:  As is stated, the number of ewes was not felt adequate to evaluate potential effects of parity.

Reviewer 2 Report

This is a very interesting topic and I commend the authors for the collection of all of these various data points, as there is large amount of information in the paper. Overall, I believe this is a paper with a lot of merit and very interesting results looking at different hair sheep breeds.

I do have a few general project wide questions that I will start with and then some specific edits within the manuscript itself.

- First, what was the rationale or thought process behind the division of periods utilized, especially 1 -5?

- Also, period 6 & 7 were from the end of supplementation until lambing correct, and if so why were they separated?

- I know there was a limited number of multiparous ewes which did not allow for separate analysis, but was an analysis done using only the primiparous ewes to determine if any of the overall results changed?

Specific items:

- Line 29 & 30, was the average age of all ewes 2.8 yrs or just the primiparous ewes?

- Line 34 - 40, this is a long sentence with all of the data in it and it might benefit from removing the data or breaking it into separate sentences

- Line 40 & 47, there is either KAT-Low or DOR-Low as opposed to "KAT-LS" or "DOR-LS"

- Line 121 needs a unit following the second 3.66

- Line 129 How exactly were ewes weighed? Once on full feed? Once following "X" hrs feed restriction? Two consecutive days? etc.

- Line 140 Van Soest reference should be numbered and added to References

- In Table 1: ADF of the supplements was not mentioned in Materials & Methods

- Tables 2 & 8 are very challenging to read as they are presented. They may benefit from being presented as Landscape tables or it may just be difficulty within the proof itself.

- Tables 4 & 5 have some spacing issues, specifically within the heading of "Brd*Sup" p-value and within the "Brd" column of main effects

- In Table 6 there is a missing ")" at the end of the HE:HR

- Lines 394 & 395 Kaufmann & Machado references need numbering and added to References

- Be sure to double check references for completeness and correct numbering given the additional references listed previously

Author Response

This is a very interesting topic and I commend the authors for the collection of all of these various data points, as there is large amount of information in the paper. Overall, I believe this is a paper with a lot of merit and very interesting results looking at different hair sheep breeds.

I do have a few general project wide questions that I will start with and then some specific edits within the manuscript itself.

Response: Thanks for your good suggestions and corrections.

- First, what was the rationale or thought process behind the division of periods utilized, especially 1 -5?

Response: Although mean values or changes in total periods 1 to 5 give an idea about overall effect of treatment in entire period, it cannot capture dynamic changes affected by treatment in different periods. For example, wheat straw intake was not affected by supplement treatment in period 1 and 2, but subsequently affected by supplement treatment. We attempted to explain this response in the manuscript.

- Also, period 6 & 7 were from the end of supplementation until lambing correct, and if so why were they separated?

Response: Yes, it is correct that periods 6 and 7 were from the end of supplementation until lambing. As explained earlier, we separated period 6 and 7 to understand the changes in the last two months of pregnancy affected by previous nutritional supplements. We have explained the rationale of using different period in the revised manuscript.

- I know there was a limited number of multiparous ewes which did not allow for separate analysis, but was an analysis done using only the primiparous ewes to determine if any of the overall results changed?

Response:  It is a good suggestion. It is not common for sheep flocks to consist only of primiparous animals, which is a consideration for inclusion of ewes.  However, analysis of some variables with only ewe lambs was conducted, with similar findings.  An example is litter birth weight, with P values of 0.168, 0.0499, and 0.160 for Brd, Sup, and BrdxSup.

Specific items:

- Line 29 & 30, was the average age of all ewes 2.8 yrs or just the primiparous ewes?

Response: It was the average age of all ewes.

- Line 34 - 40, this is a long sentence with all of the data in it and it might benefit from removing the data or breaking it into separate sentences.

Response: we have made two separate sentences and improved the clarity.

- Line 40 & 47, there is either KAT-Low or DOR-Low as opposed to "KAT-LS" or "DOR-LS"

Response: Thanks. We have corrected now.

- Line 121 needs a unit following the second 3.66

Response: Thanks. The unit has been now provided.

- Line 129 How exactly were ewes weighed? Once on full feed? Once following "X" hrs feed restriction? Two consecutive days? etc.

Response: This has been added.  They were weighed full, with a shrink period, before removing refusals and dispensing new feed.

- Line 140 Van Soest reference should be numbered and added to References

Response: It was a mistake. The number has been added.

- In Table 1: ADF of the supplements was not mentioned in Materials & Methods

Response: Thanks. The analysis method of ADF in supplement has now been mentioned.

- Tables 2 & 8 are very challenging to read as they are presented. They may benefit from being presented as Landscape tables or it may just be difficulty within the proof itself.

Response: Originally it was in landscape, but during formatting by the editorial office as per the style of this journal, it was changed. We have now made it better and final formatting will be done in the proof.

- Tables 4 & 5 have some spacing issues, specifically within the heading of "Brd*Sup" p-value and within the "Brd" column of main effects

Response: We have corrected it though originally submitted manuscript had not this problem. It was changed when the manuscript underwent formatting as per the styles of this journal.

- In Table 6 there is a missing ")" at the end of the HE:HR

Response: It has been corrected now.

- Lines 394 & 395 Kaufmann & Machado references need numbering and added to References

Response: Thanks. We have corrected it

- Be sure to double check references for completeness and correct numbering given the additional references listed previously.

Response: we have checked further.

Round 2

Reviewer 2 Report

Tables are much easier to read now and I fully understand that journal formatting played a part in their difficulty to read in Version #1.

I understand that the periods were used to identify dynamic changes caused by treatment, but what is the rationale for the specific breakpoints used to determine period 2 vs. 3 or 6 vs. 7 etc.?

- Line 122 & 123: suggest changing sentence to read "treatments were randomly assigned, using a random number table, to the groups (Figure 1)."

Author Response

Tables are much easier to read now and I fully understand that journal formatting played a part in their difficulty to read in Version #1.

Response: Thanks that it is now better.

I understand that the periods were used to identify dynamic changes caused by treatment, but what is the rationale for the specific breakpoints used to determine period 2 vs. 3 or 6 vs. 7 etc.?

Response: we have explained gain more specific way in the texts.

- Line 122 & 123: suggest changing sentence to read "treatments were randomly assigned, using a random number table, to the groups (Figure 1)."

Response: Thanks, we have revised the sentence.
